# Using Electroporation to Improve and Accelerate Zebrafish Embryo Toxicity Testing

**DOI:** 10.3390/mi15010049

**Published:** 2023-12-26

**Authors:** Nusrat Tazin, Tamara J. Stevenson, Joshua L. Bonkowsky, Bruce K. Gale

**Affiliations:** 1Department of Electrical and Computer Engineering, University of Utah, Salt Lake City, UT 84112, USA; 2Department of Pediatrics, University of Utah School of Medicine, Salt Lake City, UT 84112, USA; 3Department of Mechanical Engineering, University of Utah, Salt Lake City, UT 84112, USA

**Keywords:** toxicity testing, microinjection, electroporation, zebrafish embryo, nanoparticles

## Abstract

Zebrafish have emerged as a useful model for biomedical research and have been used in environmental toxicology studies. However, the presence of the chorion during the embryo stage limits cellular exposure to toxic elements and creates the possibility of a false-negative or reduced sensitivity in fish embryo toxicity testing (FET). This paper presents the use of electroporation as a technique to improve the delivery of toxic elements inside the chorion, increasing the exposure level of the toxins at an early embryo stage (<3 h post-fertilization). A custom-made electroporation device with the required electrical circuitry has been developed to position embryos between electrodes that provide electrical pulses to expedite the entry of molecules inside the chorion. The optimized parameters facilitate material entering into the chorion without affecting the survival rate of the embryos. The effectiveness of the electroporation system is demonstrated using Trypan blue dye and gold nanoparticles (AuNPs, 20–40 nm). Our results demonstrate the feasibility of controlling the concentration of dye and nanoparticles delivered inside the chorion by optimizing the electrical parameters, including pulse width, pulse number, and amplitude. Next, we tested silver nanoparticles (AgNPs, 10 nm), a commonly used toxin that can lower mortality, affect heart rate, and cause phenotypic defects. We found that electroporation of AgNPs reduces the exposure time required for toxicity testing from 4 days to hours. Electroporation for FET can provide rapid entry of potential toxins into zebrafish embryos, reducing the time required for toxicity testing and drug delivery experiments.

## 1. Introduction

The zebrafish model is of importance in toxicology research and is frequently used for in vivo testing and determining the toxicity of chemicals in the embryonic stages of fish [1,2]. Zebrafish share 70% of genomic material with mammals, making the model useful for drug screening, human disease modeling, and chemical toxicity screening [3]. Other benefits of zebrafish include low husbandry costs compared to other vertebrate models such as rodents, pigs, etc. [3]. Also, compared to other fish models (such as trout), zebrafish adults have a smaller size (1–1.5 inches), require less space, have rapid early development, and have a high fertility rate [4]. Zebrafish remain transparent throughout their development from embryo to adult stage, making the model suitable for chemical exposure screening [5]. The adverse effect of chemicals on the development of the heart, brain, size, and other organs can be monitored in an ongoing fashion with a simple dissecting microscope without killing or dissecting the organism [5]. Moreover, zebrafish have the potential to recover their organs from damage that occurs during the development process and can be used for large-scale screening (>1000 embryos [6]).

To reduce suffering and pain caused by acute toxicity testing on adult fish, an alternative is early embryonic stage toxicity testing. In the fish embryo acute toxicity (FET) test, newly fertilized embryos are exposed to chemicals for 96 h. Toxicity is determined based on deformity observation. FET is a powerful method to mimic maternal pollutant transfer to the offspring by exposing the early-stage embryos to pollutants and shortening the required experimentation time. Lammer and his group [7] showed a statistical correlation between fish embryo tests and acute fish toxicity tests, which indicates the acceptability of early-stage fish testing. However, some significant differences have been observed for substances with higher molecular weights for embryo vs. adult toxicity testing [8]. One primary cause of the difference in toxic effects is determined by the presence of an acellular membrane known as the chorion that surrounds early-stage embryos. This chorion is made of fibrillar proteins and glycoproteins that form intermediate and outermost layers with some nanometer pore canals [9], but flow in and out of the chorion is significantly reduced. These pores are different than the micrometer-sized canals known as micropyles for the entry of sperm that close after egg fertilization [10,11]. It is still unclear whether this envelope protects embryos from exposure to certain chemicals. Reportedly, the small pore size restricts the uptake of materials smaller than the pore diameter [12]. Moreover, polymers and higher-molecular-weight surfactants are expected to be blocked by the protective layer. Limited evidence has been found on the hardening of the chorion during development, changing the permeability even for smaller molecules [13]. Because of the unclear impact of the chorion, researchers often remove the chorion by trypsin/EDTA [3] or pronase [2] or mechanically by forceps [2]. Experiments that remove the chorion before 24 h post-fertilization (hpf) lead to low survival [5].

Since the presence of the chorion at early stages creates a potential risk of generating false-negative results in toxicity studies due to reduced permeability [14], there is a need for methods to enable the chemicals being tested to be transported inside the chorion of embryos so that any toxic effects can take effect. Moreover, the FET test requires a minimum of four days of exposure to the chemicals before determining any noticeable phenotypical changes in the embryos, making the process time-consuming and labor-intensive. Schuber and his lab [15] implemented microinjections to expose substances inside the chorion for zebrafish embryos. However, it is a low-throughput process, and injecting one embryo at a time is laborious and time-consuming. Mandrell et al. [6] developed an automated chorion removal system for toxicity. However, the system is expensive and uses an enzymatic process for chorion removal, whose adverse effects cannot be avoided.

Electroporation is a potential technique for helping to open up the chorion and allow chemicals and particles to enter. Electroporation is a technique where short electric pulses are provided to break down the cell membrane’s dielectric layer, which creates multiple pores in the membrane, allowing particles and molecules to enter. Electroporation is often used to enable the entry of foreign DNA into a cell [16] and is thus used for gene transfection, tumor treatment [17], and cell-based therapies [18]. Though this method has also been implemented for the transfection of zebrafish embryos [19,20], the efficiency is not comparable with conventional microinjection methods [21]. However, electroporation as a method in toxicology research could be beneficial but is yet to be explored. By optimizing parameters like the number of electric pulses and their duration, this method could assist in delivering material inside the chorion for early-stage exposure of embryos and determining the toxicity effect for zebrafish without removing the chorion and could reduce the probability of false-negative outcomes. 

In this paper, an electroporation system has been designed and developed that is appropriate for early-stage (0–48 h post-fertilization) zebrafish embryos for fish toxicity testing. The system has been implemented to determine the effectiveness of electroporation as a novel method in FET without removing the protective chorion layer to reduce the risk of false-negative results and the required testing time. Trypan blue dye, gold nanoparticles (AuNPs), and silver nanoparticles (AgNPs) have been used as electroporation delivery material. Testing with Trypan blue dye and AuNPs indicated that different concentrations of material could be delivered inside the chorion of the embryos by varying the electric field. Optimizing the pulse parameters can result in the diffusion of material inside the chorion with >80% survival of the embryos. A phenotypical deformity study using AgNPs showed that with electroporation, a similar deformity rate could be achieved for toxin exposure in minutes rather than days, like in the conventional process. To assess phenotypical deformity, heart rate in beats per minute, survival rate, and deformity of any body parts were considered. The system provided repeatable results for electroporation, indicating the feasibility of the method for toxicity studies.

## 2. Materials and Methods

### 2.1. Ethics Statement

Experiments with zebrafish embryos were performed following the guidelines of the University of Utah Institutional Animal Care and Use Committee (IACUC), regulated under federal law (The Animal Welfare Act and Public Health Services Regulation Act) by the U.S. Department of Agriculture and the Office of Laboratory Animal welfare at NIH.

### 2.2. Fish Stock and Embryo Raising

Adult fish were bred according to the standard protocol. After electroporation, the embryos were kept at 28 °C in E3 medium for further observation [22].

### 2.3. Device Theory

Electroporation is a method that effectively introduces DNA, RNA, or proteins into bacteria, yeast, and mammalian cells. This method is also used for gene transfer in fish embryos and could be made feasible by altering pulses’ voltage amplitude and frequency [23,24]. High field strengths and long durations may damage the cell irreversibly, which is why optimization of the parameters is necessary [25]. Here, we are using electroporation as a method for delivery of toxic material inside chorion for zebrafish embryos. However, for electroporation with fish embryos, the characteristic protective chorion must also be considered when determining the electrical parameters. For zebrafish embryos, reports using electroporation [19,26] for gene transfer indicated the requirement of removal of the chorion or insertion of genetic material inside chorion for success [27]. The chorion already contains nanometer-scale pores [28], and the effect of the electric field on the chorion is not well understood. To better understand the effect of electroporation on the chorion, this work will vary the pulse parameters for electroporation to determine how they impact the pore size of the chorion without affecting the embryos inside the chorion. 

### 2.4. Device Design and Operation

The electroporation system developed in this work consists of an electroporation chip and an electroporation circuit for providing electric pulses. The electroporation chip is a microfluidic chip that contains microscale zebrafish embryos and media. The schematic and the actual connections of the electric circuit can be seen in Appendix A, respectively. The circuit consists of a DC source (Keithley DC Power Supplies, Tektronix, Beaverton, OR, USA), an Arduino Uno (Arduino LLC, Ivera, Italy), and a driver circuit (L298N, STMicroelectronics, Geneva, Switzerland). The MOSFET inside the driver circuit switches pulses from the DC source to the electrodes in a pattern of on/off pulses programmed on the Arduino Uno. The circuit can provide voltages from 5 to 25 V and different positive and negative pulses. More details can be found in [27].

The electroporation chip consists of a chamber for holding the embryos and the media that need to be delivered. The embryo size is 700 μm, and the chamber can hold up to 200 µL of media after holding 20 zebrafish embryos at 0 h post-fertilization (hpf) inside the chamber. The chamber size was 10 mm × 3 mm × 10 mm. The chamber is created by curing a PDMS layer (Sylgard 184, Dow Corning, Midland, MI, USA) in a glass Petri dish and cutting away the middle part of the layer with a sharp razor. Two platinum electrodes were inserted on both sides of the chamber, connecting to the circuit to provide pulses. The dimensions of the platinum electrodes are 10 mm × 10 mm × 0.1 mm (Newvision1981, USA) with a gold-plated copper handle. The schematic and image of the electroporation chip is presented in Figure 1. Figure 1a shows the setup schematic for an electroporation chip. Figure 1b shows the image of the actual electroporation chip. 

With the electroporation chip and circuit assembled, we determined the electric parameters feasible for the embryos of age 0–1 hpf with a high survival rate. We applied voltages from 0 to 25 V with a different number of pulses (2–10) and pulse duration (15–100 ms). The goal was to determine voltage amplitude and pulse parameters that would not affect development or survival. Empirically, voltages from 5 to 20 V with 15 ms pulse duration and four positive pulses with six alternating positive and negative pulses work best for our toxicity testing regarding survivability and fish embryo toxicity testing. The oscilloscope image of the voltage pulses can be seen in Appendix A.

In the device operation protocol, approximately 20 embryos were loaded into the chip chamber using a standard transfer pipette from a Petri dish containing E3 media. The age of the embryos was 0–1 h post-fertilization (hpf). After that, the E3 media was removed from the chip holder with another standard pipette. Care must be taken during this process to avoid damaging the embryos. The chamber was filled with 100 µL of the toxin material. The electrodes were provided with electric pulses of different amplitudes and durations for electroporation. The embryos were kept inside the chamber for some time to distribute the toxin material inside the chorion. After that, the embryos were removed from the chamber and washed three times with E3 media before imaging. In the final step, embryos were kept in a fresh E3 media and incubated for further observation. The process flow can be seen in Figure 2. Figure 2-Step 1 shows the collection of embryos with a standard pipette from a Petri dish. Figure 2-Step 2 shows the embryos’ placement in the electroporation system’s electroporation chamber. After this step, the E3 media from the chamber was removed using a standard pipette. Figure 2-Step 3 shows the chamber filled with the toxic solution or the dye we were testing. After this step, the electroporation circuit provided the platinum electrodes with voltage pulses. Then, the embryos were removed from the EP chamber and washed three times with fresh E3 media before placing them in a new Petri dish with fresh E3 media. This step is shown in Figure 2-Step 4. After waiting for 1 h, images were taken, and embryos were incubated at 32 °C for further development. 

### 2.5. Toxicity Analysis Method

#### 2.5.1. Dye and Nanoparticle Preparation

Trypan blue dye (0.4% Gibco, Grand Island, NY, USA), 20 nm gold nanoparticles (AuNPs) (Sigma-Aldrich, St. Louis, MO, USA), and silver nanoparticles (AgNPs) of 10 nm particle size (Sigma-Aldrich) were used for toxicity testing with the electroporation system. Trypan blue is not toxic to zebrafish embryos but was used to determine the effect of the electric field. The concentration of the nanoparticles was 0.02 mg/mL, and the aqueous buffer contained sodium citrate as a stabilizer. The 40–100 µg/mL AgNP solutions were prepared from the 0.02 mg/mL concentration of AgNPs by centrifugation. To enhance stability and reduce agglomeration, we sonicated the concentrated AgNPs before performing experiments with zebrafish embryos. 

#### 2.5.2. Analysis with Trypan Blue Dye

Trypan blue dye was used in various concentrations to determine whether the electric pulses affected the delivery of materials inside the chorion. A calibration curve was generated by diluting the Trypan blue dye in various concentrations and exposing the 0 hpf embryos for 5 h. The existing pores and the small size of the dye particles allow the natural diffusion of dye inside the chorion with time. The embryos were washed three times before imaging with a Dinolite microscope. ImageJ 1.53p software was used to determine the RGB value of the images of the exposed embryos, taking one embryo at a time and calculating absorbance using the following equation [29].
(1)$$A_{R/G/B}=-{log}_{10}\left(\frac{R_{s}/G_{s}/B_{s}}{R_{c}/G_{c}/B_{c}}\right)$$

The concentration of the dyes inside the chorion can be easily calculated from the original dye’s known concentration value and the generated calibration curve. The calibration curve is presented in Appendix A. The initial concentration of the Trypan blue was 0.4 mg/mL. Three concentrations were generated by diluting with E3 media (0.04, 0.004, 0.0004 mg/mL). For each concentration, *n* = 6, absorbance data were collected with the RGB values of the images of the embryos. For these concentrations, the calibration curve provided R^2^ = 0.97 and equation $y=0.1147\ln\left(x\right)+0.8559$. From this equation, the corresponding concentration of Trypan blue was calculated with the known absorbance value. Appendix A shows the average absorbance of the different Trypan blue exposed embryo concentrations. 

To determine the effect of electroporation, after exposing the embryos to the dyes in the chamber, electric pulses of 5–20 V were applied with ten positive/negative alternating pulses of pulse duration 15 ms with a 50 ms pulse gap. The pulse parameters were determined empirically. After exposure of 10 min, the embryos were removed from the chamber and washed three times. The embryos were kept in fresh media for 1 h to distribute the inserted dyes throughout the chorion properly before taking images. The images of the embryos were then further used to calculate the concentration and absorbance of Trypan blue inside the chorion. For control, embryos were exposed to Trypan blue dye inside the EP chamber for 10 min without applying an electric field. 

#### 2.5.3. Analysis with Nanoparticles

AuNPs were used to determine the approximate pore size present in the chorion. Moreover, with AuNPs and AgNPs, we checked the concentration variation inside the chorion caused by applying an electric field similar to the Trypan blue experiments in Section 2.5.2. After applying voltages, we determined the phenotypical changes to the developed embryos from the toxic effect of AgNPs. In other words, we determined whether the toxic effect of the AgNPs was increased due to electroporation.

##### Determining the Pore Size of the Chorion with Nanoparticles

Estimating the pore sizes in the chorion of the embryos was needed in order to estimate the size of the nanoparticle that would pass through the chorion. AuNPs ranging in size from 20 to 240 nm were used to make this determination. The embryos were soaked in AuNPs of various sizes for 5 h. After 5 h, images of the embryos were taken. The AuNPs are red and visible to the eye. However, to determine the exact entry of the particles, absorbance was calculated at a wavelength of 650 nm for AuNPs using the RGB value of the images with Equation (1) using the software ImageJ. The calculated value of the absorbance determined whether the nanoparticles had been inserted into the chorion in the diffusion process. As a control, embryos without any exposure to the AuNPs were also observed. 

##### Determining the Absorbance after Electroporation

To determine how many nanoparticles diffuse inside the chorion due to the application of pulses, absorbance calculations similar to the Trypan blue process were performed. Around 20 embryos were placed inside the chamber of the chip. Ten alternating positive/negative pulses with an amplitude of 8 V and a duration of 15 ms with a 50 ms pulse gap were applied. After that, embryos were kept inside the chamber for 20 min to allow particles to diffuse inside the chorion. The embryos were taken out, washed, and kept in a Petri dish with fresh E3 media. After 1 h, images were taken for further analysis. Three controls were considered: Control 1: Embryos not exposed to nanoparticles or voltage pulses. Control 2: Embryos exposed to voltage pulses only to determine their effect on survivability. Control 3: Embryos exposed to nanoparticles only. This test will help to determine whether the application of voltage will enable more diffusion of particles inside the chorion. The process was performed for both AuNPs and AgNPs. From the images, the RGB value was calculated using ImageJ. The absorbance was calculated at 650 nm for 20 nm AuNPs and 420 nm for 10 nm AgNPs. For each data point, *n* = 6. The concentration of AuNPs and AgNPs was 0.02 mg/mL. 

##### Determining the Toxic Effect on the Phenotypes of Embryos

From the literature, it was found that the AuNPs had a minimal effect on the phenotypes of zebrafish embryos when a very high concentration (100 µg/mL) was used [30]. No deformity was observed for lower concentrations (<100 µg/mL). Therefore, we used 10 nm AgNPs (Ag-NP) and avoided using AuNPs to determine the effect on phenotypes of the zebrafish embryos after exposure to early stages. The literature [27] showed that AgNPs affect the survival rate, heart rate, and physical deformity of the zebrafish embryos, and the effect increases with increasing concentration. The significant phenotypical changes start to show at a concentration of 40 µg/mL and higher, which is why various concentrations of AgNPs from 20–100 µg/mL were prepared for the control experiments. 

For control experiments, the embryos were exposed to AgNP concentrations of 20–100 µg/mL for 24 h, followed by measurement of deformity rate, deformity type, and survival rate. That provided a baseline to determine what deformity type and rate to expect when embryos are exposed to AgNPs of various concentrations. After 5 h of exposure, similar to the previous absorbance calculation, we determined the absorbance for various concentrations from the RGB values of the images taken. Survival and deformity rates were monitored after 48 hpf when the embryos were out from the chorion. Any organ’s abnormal growth, tail tilting, or deforming body shapes were considered as deformity. This step is undertaken to replicate the actual FET test and determine the deformity rate to have a baseline for further testing.

For determining the effect of electric field application on nanoparticle exposure, 60 µg/mL, 10 nm AgNPs were considered. After the embryos were suspended, the chamber was filled with 60 µg/mL of AgNPs, which was followed by an application of ten alternating positive/negative pulses of 8 V and 15 ms duration and 50 ms gap. After voltage application, the embryos were kept inside the chamber to allow the nanoparticles inside the chorion to diffuse properly. After transferring the embryos to the new media, they were incubated for 48 h. At 72 hpf, the deformity, survival, and heat rates were determined and compared to controls. For determining heart rate, a Dinolite (AF4115ZT) USB microscope (Dunwell Tech, Inc., Torrance, CA, USA) was used to capture video of each embryo, and the heart rate per minute was calculated by visual inspection.

#### 2.5.4. Survival and Morphological Analysis

Survival and morphological analysis were conducted under a dissecting microscope at 24 hpf. The animals were considered to have survived if their heartbeat was observed. 

#### 2.5.5. Statistical Analysis

Statistical analysis was performed using JMP 15 (SAS), and a *t*-test was performed for a two-way comparison. ANOVA with post hoc Tukey’s HSD between individual means was performed for multiple group and interaction profile comparison.

## 3. Results and Discussion

### 3.1. Analysis with Trypan Blue Dye

Trypan blue dye was used to quantify the amount of small molecules that enter the chorion during electroporation. Figure 3 shows the concentrations and measures of optical absorbance of the Trypan inside the chorion after applying various voltages, keeping the pulse duration, the number of pulses, and the pulse gap fixed. In Figure 3a, it can be observed that the average concentration of blue dye for control embryos is 0.91 µg/mL, and the corresponding absorbance is 0.05, as shown in Figure 3b. For 5 V, the average concentration is 2.35 µg/mL, and for 20 V, it is 5.74 µg/mL. Similarly, absorbance values for 5 V and 20 V are 0.17 and 0.26, respectively. These results indicate that with increasing voltage, the Trypan blue concentration inside the chorion increases, as indicated by an increase in absorbance value. The student *t*-test shows a significant difference between the different levels of voltage and control (*p*-value < 0.05). The student *t*-test result can be seen in Appendix A. This result confirms that different concentrations of Trypan blue dye can be inserted into the chorion by varying the electric field. However, the survival rate for the application of a voltage of more than 10 V was less than 50% compared to the control. Therefore, for later testing, voltages lower than 10 V were considered. The survival data are presented in Appendix A. The intensity of the Trypan blue for different voltages can be observed in Figure 3b, which shows the relative intensity increase of the dye color with the application of higher voltages. 

### 3.2. Analysis with Nanoparticles

#### 3.2.1. Determining the Native Transport Rate of Nanoparticles

At an early stage, the chorion layer around the embryo already contains some poorly characterized pores. To determine the size of the nanoparticles for toxicity testing, we needed to determine the effective pore sizes in the chorion. We used AuNPs from 20 to 240 nm. After soaking the embryos in nanoparticles of different sizes for 5 h, we observed that nanoparticles 40 nm and larger could not pass through the chorion. However, 20 nm particles naturally diffuse inside the chorion. The particle intensity for various sizes can be observed in Figure 4. Here, we can see from Figure 4a that the inside of the chorion is filled with 20 nm AuNPs. The purple tint around the chorion in Figure 4b indicates that nanoparticles are present around the chorion layer but could not penetrate the pores. Figure 4c shows a control embryo not exposed to the AuNPs. The absorbance for embryos exposed to 20 nm nanoparticles is 0.34, and for 40 nm, it is 0.047. Statistically, there is no significant difference between the control absorbance and absorbance of the 40 nm particles (*p*-value > 0.05), indicating negligible entry of nanoparticles to the chorion. From this analysis, it seems likely that the chorion pores are less than 40 nm in size but larger than 20 nm. Thus, nanoparticles < 20 nm will be used to test the electroporation hypothesis for toxicity testing. 

#### 3.2.2. Determining Absorbance after Electroporation

Similar to the Trypan blue absorbance study, to determine whether electroporation can insert more nanoparticles inside the chorion or not, we have used 20 nm AuNPs and 10 nm AgNPs. Figure 5 shows the absorbance measured for various conditions and provides some representative intensity images for AgNPs and AuNPs. The raw data are in Appendix A. Figure 5a shows images of embryos exposed to 8 V pulses and 10 nm AgNPs. Figure 5b shows embryos only exposed to the AgNPs but not to any electric field. The images clearly show the higher intensity of AgNPs inside the chorion after voltage application. Figure 5c shows the embryos exposed to AuNPs for only 15 min. Figure 5d shows embryos exposed to AuNPs for 15 min after application of 8 V pulses. For each condition, *n* = 6. A similar gold intensity difference can be observed here for embryos exposed to voltage pulses and not exposed to pulses. The figures show that the absorbance value is much higher when the embryos are exposed to AgNPs with 8 V pulses than when they are only exposed to AgNPs. The average absorbance was 0.04 and 0.01, respectively, making the electroporated value more than three times the control value.

Moreover, a similar situation can be observed for the AuNPs. The average absorbance when embryos were only exposed to the AuNPs is 0.03; when exposed to electroporation, the absorbance is 0.04, ~1.5 times the control. Figure 5e shows the absorbance for the mentioned conditions. The particle size of AgNPs (10 nm) is smaller than AuNPs (20 nm). That is why, after the application of electrical pulses, the absorbance value is much higher for AgNPs compared to the AuNPs. This confirms that the electric field aids the delivery of toxic material inside the chorion. From the *t*-test, the *p*-value of the absorbance of AgNP was 0.0017. This indicates that there is a statistically significant difference between the absorbance of only AgNP inserted inside the chorion by diffusion and that of AgNP inserted due to voltage application. Similarly, for AuNP, the *p* value is 0.005 and indicates statistical significance. The statistical significance is presented in Appendix A and Figure 5e. 

#### 3.2.3. Determining the Toxic Effects Using Embryo Phenotypes

##### Control Experiment Results 

A series of experiments were performed to determine how electroporated embryos responded to exposure to toxic materials. For the control experiment, embryos were exposed to AgNPs with concentrations of 20–100 µg/mL for 24 h. Appendix A shows the different concentrations of AgNPs inside the chorion after 5 h of exposure. Figure 6 and Appendix A present the calculated absorbance for different concentrations. Figure 6a shows a linear absorbance increase for increasing silver nanoparticle concentrations. Phenotypic deformity was measured after 48 hpf when embryos were out of the chorion. The noted deformities include undeveloped body parts, a deformed body shape, or a tilted tail. The deformity and survival rate associated with various concentrations of nanoparticle exposure for 24 hpf is presented in Figure 6b and Appendix A. The figure shows that control embryos have a survival rate of >80% and a deformity rate of 0%. However, the survival rate decreases, and the deformity rate increases with increasing nanoparticle concentration. No deformity was observed at 20 µg/mL concentration, but the survival rate decreased relative to the control. Deformity starts to show from 40 µg/mL, and at 100 µg/mL, it reaches 80%, meaning that out of the 25% surviving embryos at 100 µg/mL, 80% were deformed. The embryos were not exposed to any voltage pulses in these experiments. 

##### Deformity Rate with Electric Field Application

From the control experiment results presented in Section “Control Experiment Results”, it can be seen that a 20 µg/mL Ag nanoparticle concentration does not correlate with any deformity to the embryos, and 100 µg/mL affects nearly all embryos severely. Accordingly, in order to have a range that was somewhat sensitive for testing with electroporation, we chose 60 µg/mL, which gave a 33.3% deformity rate in control experiments. The goal was to determine whether exposing embryos to 60 µg/mL for 40 min with voltage pulses would provide a similar deformity percentage to any control experiments. 

Figure 7a shows the effect of electroporation on absorbance for embryos exposed to 60 µg/mL AgNPs for 40 min. The electroporated embryos had absorbance nine times that of those exposed only to nanoparticles. 

Figure 7b,c show the effect of electroporation on deformity rate with exposure of 10 nm AgNPs for 40 min. The experiment is repeated five times with a different batch of embryos. The number of embryos used in the experiments is presented in Appendix A. Here, control embryos are not exposed to any nanoparticles or voltage pulses. The deformity rate data for five repetitions are presented in Figure 7b and Appendix A. Figure 7c shows the average deformity rate for five repetitions. The deformity percentage after voltage application with nanoparticles was around 32.2% (Figure 7c, Nano+V). This is similar to the deformity rate of control experiments with 60 µg/mL nanoparticle exposure presented in Section “Control Experiment Results”. Here (Figure 7b), embryos exposed to the nanoparticles only (Only Nano) showed no deformity except for one experiment, possibly due to a bad clutch. Moreover, deformity is zero for embryos only exposed to voltage pulses (Only V) on all repeats except one (Figure 7b). This result indicates that voltage application does not increase the embryos’ deformity rate. After performing the *t*-test on the five repeats of the data presented in Figure 7b,c, it was found that the *p*-value is <0.05 when Nano+V is compared with Control 1 (No V+ No particle). However, *p* is >0.05 when Only V and Only Nano is compared with Control 1 (No V+ No particle). The student *t*-test data can be seen in Appendix A. This analysis indicates that the deformity rate for voltage application with nanoparticles is statistically significant when compared with Control 1. But for only exposure to nanoparticles or voltage, the deformity rate of five repeats is not statistically significant. 

Therefore, from repeated results, it can be said that the deformity rate for 40 min of exposure to the electric field is similar to the rate when exposed for 24 h without any external field (shown in Figure 6b) for a 60 µg/mL concentration. When exposed to nanoparticles for 24 h after the application of voltage pulses, the deformity rate was much higher, and the survival rate was close to zero. However, multiple repetitions are required to find statistical significance for that conclusion. 

The data in Figure 7 indicate that the application of an electric field can reduce the required toxic material exposure test time from 24 h to 40 min of exposure while still obtaining the same deformity rate for the same concentration. A sample of observed deformities observed in these experiments is presented in Figure 8. 

##### Condition of Heart Rate after Electric Field Application

Heart rate is another parameter affected by the toxicity of nanoparticles. The heart rate of zebrafish embryos upon exposure to 60 µg/mL of AgNPs (*n* = 6) can be seen in Figure 9. This shows that the heart rate is lower when embryos are exposed to nanoparticles and electric fields than for other conditions. The full heart rate data can be seen in Appendix A. The data indicate that the nanoparticles or the electric field alone do not affect the heart condition significantly. With both Ag NP exposure and electroporation, more nanoparticles are inserted into the chorion, mimicking the condition where no chorion is present. After statistical analysis, it was found that there is a significant difference between the heart rate condition of the control embryos and embryos exposed to nanoparticles with voltage pulses represented as a ‘*’ in the figure (*p*-value < 0.05), but for other conditions, a significant difference was not found statistically. The *p*-value data are presented in Appendix A.

##### Survival Rate after Electric Field Application

The survival rate of the embryos after exposure to 10 nm AgNPs with and without the application of voltage pulses is provided in Figure 10, averaging data from five repetitions. The number of embryos used in different repetitions is provided in Appendix A. This figure shows that the survival rate for embryos exposed to nanoparticles and voltage pulses is much lower (80.25%) than the control embryos (100%). Here, control embryos are exposed to only nanoparticles. The higher survival rate of embryos exposed to only voltage (100%) indicates that the electric field alone does not affect the survival of the embryos. Moreover, the embryos only exposed to nanoparticles have a higher survival rate (91.69%) than the embryos exposed to both voltage and nanoparticles, indicating that there are not enough nanoparticles being inserted into the chorion through diffusion only to make any significant difference. The survival data are presented in Appendix A.

## 4. Conclusions

We have developed a novel zebrafish embryo toxicity testing method to solve labor-intensive and low survival issues related to removing the chorion. Moreover, this is the first report of using electroporation to help deliver toxins to the developing embryo in fish embryo toxicity testing, which opens the possibility for high-throughput toxin screening. The results of the work showed that electroporation or electric pulses can control the entry of toxic material inside the chorion so that young embryos can be exposed to the material without removing the chorion. In that way, embryo survival is significantly improved, and the laborious step of chorion removal is avoided. The approach should also reduce false-negative results caused by the chorion blocking the entry of certain toxins into the space around the embryo.

Initial testing with Trypan blue indicated that varying the electroporation voltage amplitude changes the amount of dye that enters the space inside the chorion. This result shows that toxic material entry can be controlled with the proper selection of electric parameters. Testing with AuNPs and AgNPs showed that more nanoparticles are inserted into the chorion with electric pulses. For 10 nm AgNPs, 4 times more particles passed through the chorion; for 20 nm AuNPs, it was 1.5 times higher than without any external force. These results suggest that the process of passing through the chorion is dependent on the particle size and the pore size in the chorion. One important methodological issue is that our electroporation approach is based on the application of a negative charge, so toxin testing requires negatively charged toxins.

Testing with 60 µg/mL AgNPs showed that the deformity rate with electroporation and exposure to silver for 40 min is similar to the deformity rate when embryos are exposed to only nanoparticles for 24 h without any external forces. These results suggest two things: First, due to the voltage application, more nanoparticles enter the chorion, and embryos are exposed to more toxins even when the exposure is only 40 min. Electroporation alone was shown to not affect the deformity rate. Second, with only 40 min of exposure, similar AgNP concentrations and deformity rates were observed compared to exposure for 24 h, suggesting it would be possible to reduce the FET test timing from days to hours with electroporation. Heart rate data also show that the heart rate is reduced significantly when applying both voltage and nanoparticles. Survival rates were also shown to fall significantly with exposure to both AgNPs and electroporation compared to either alone. In summary, adding electroporation with nanoparticle exposure significantly affects the heart, survival, and deformity rates. Deformity rates were similar for embryos exposed to AgNPs for a full day and embryos exposed for only 40 min with electroporation. The overall results suggest that electroporation can reduce the likelihood of false-negative results caused by the presence of the chorion without removing the chorion, open a possibility for making a high-throughput FET testing system, and reduce the experimental time from days to hours.

## Figures and Tables

**Figure 1 micromachines-15-00049-f001:**
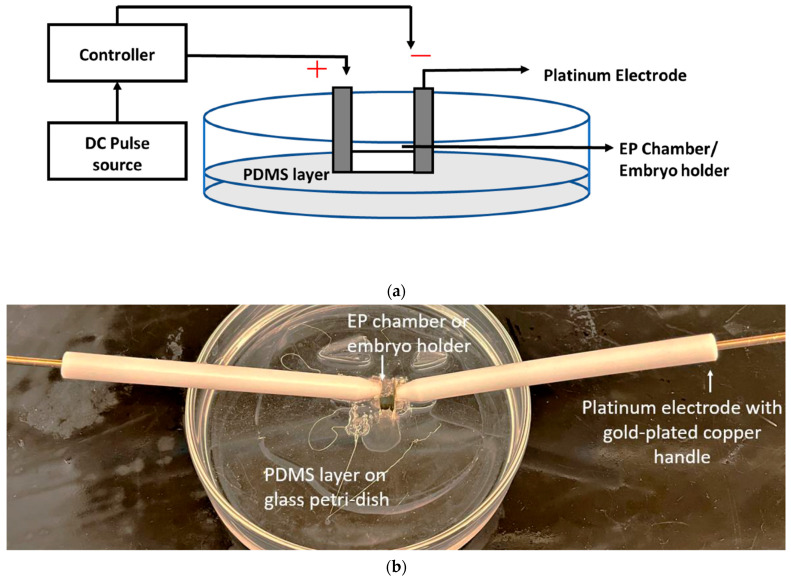
(**a**) Schematic of the electroporation setup. (**b**) Image of the electroporation chip [27].

**Figure 2 micromachines-15-00049-f002:**
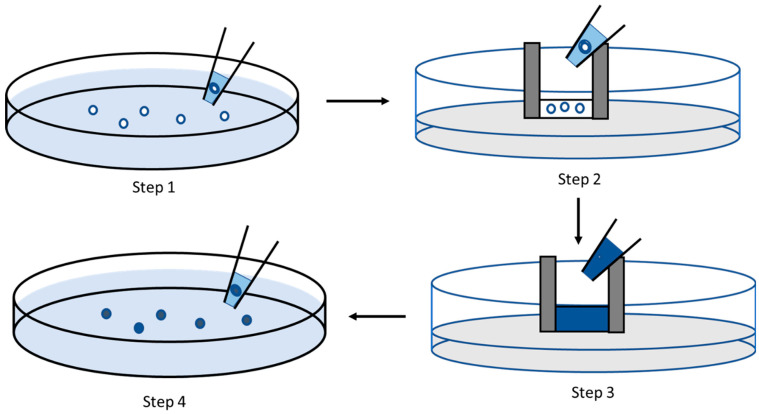
Flow diagram of using the electroporation system. Step 1: Collecting embryos from a Petri dish. Step 2: Placing the embryos in the chamber of the chip. Step 3: Applying Trypan blue or a toxic solution to the chamber and applying an electric field. Step 4: After electroporation, the embryos are transferred to a Petri dish.

**Figure 3 micromachines-15-00049-f003:**
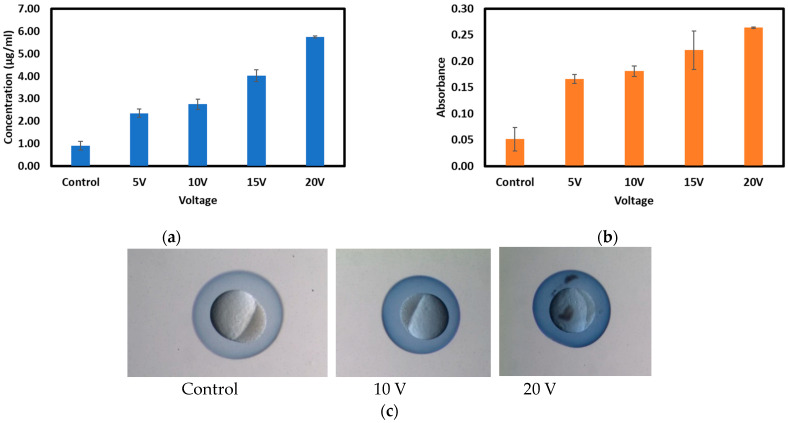
(**a**) Trypan blue concentration (µg/mL) inside the chorion for application of voltages from (5–20 V) determined from the calibration curve. (**b**) Calculated absorbance of Trypan blue inside the chorion of embryos after application of voltages from (5–20 V) determined from the RGB values of the images. (**c**) The concentration of Trypan blue inside the chorion for various conditions: Left—control embryos with no voltage application only exposed to Trypan blue dye. Middle—embryos were exposed to Trypan blue and 10 V pulses. Right—embryos exposed to Trypan blue and 20 V pulses.

**Figure 4 micromachines-15-00049-f004:**
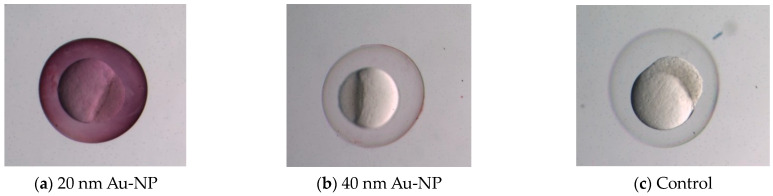
AuNPs inside the chorion of the embryos after exposure for 5 h (**a**) with AuNPs of size 20 nm (the chorion is filled with the nanoparticles) and (**b**) with AuNPs of size 40 nm. A purple tint is observed around the chorion, but no particle intensity is observed inside the chorion. (**c**) Control embryo that was not exposed to AuNPs.

**Figure 5 micromachines-15-00049-f005:**
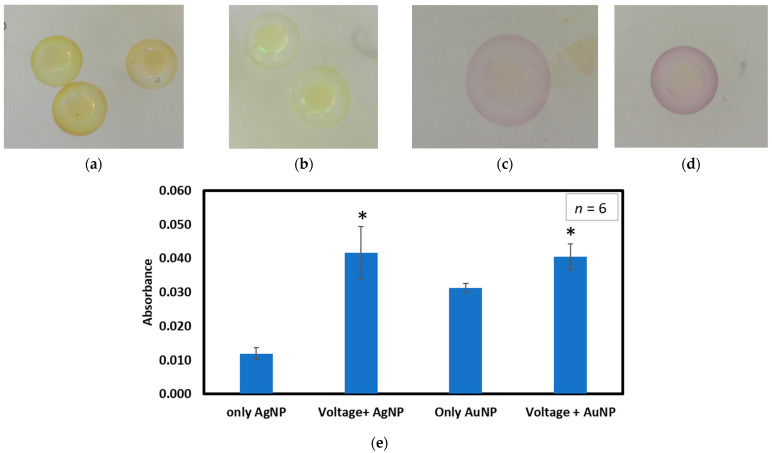
(**a**) Embryos exposed to 8 V and 10 nm AgNP for 15 min. (**b**) Embryos exposed to only AgNP for 15 min. (**c**) Embryos exposed to AuNPs for 15 min. (**d**) Embryos exposed to AuNPs for 15 min after 8V application. (**e**) Absorbance gradient for nanoparticles. The statistical significance for *p*-value < 0.05 is presented as ‘*’ in the graph.

**Figure 6 micromachines-15-00049-f006:**
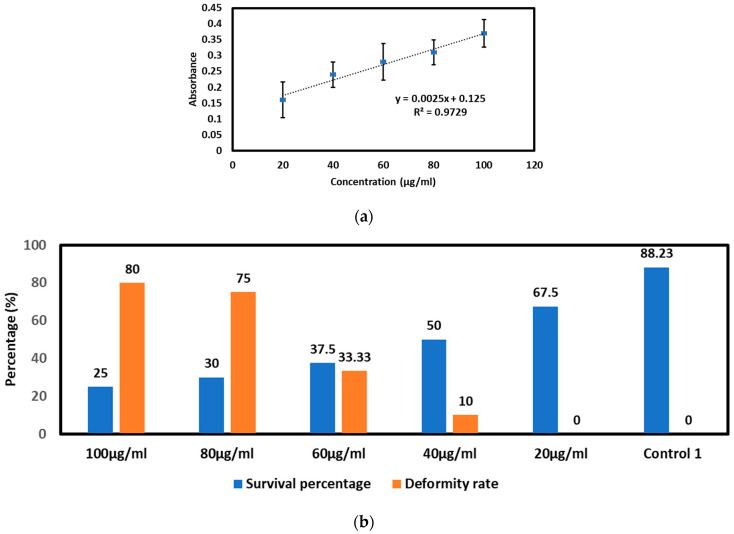
Control experiments: (**a**) Absorbance calculation for concentration 20–100 µg/mL AgNPs exposure for 5 h. (**b**) Deformity and survival rate of embryos after exposure to 20–100 µg/mL AgNPs for 24 h.

**Figure 7 micromachines-15-00049-f007:**
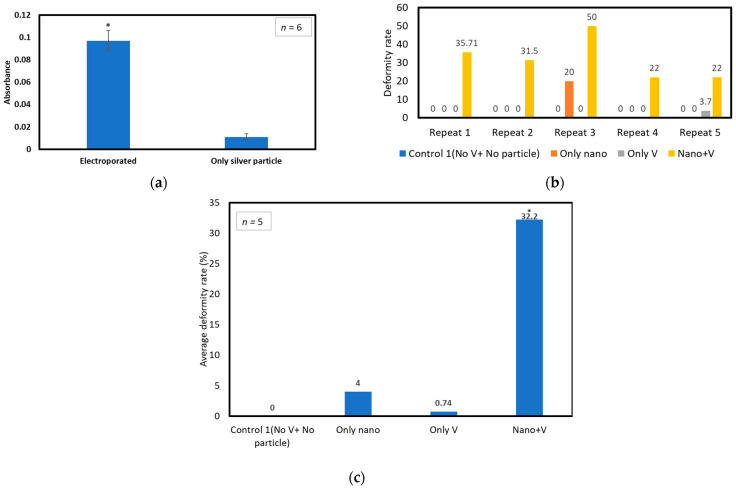
Application of electric field: (**a**) Absorbance calculation for concentration of 60 µg/mL of AgNP exposure for 40 min. “*” indicates statistically significant difference (*p*-value < 0.05) when compared with only silver particle. (**b**) Deformity rate of embryos after exposure to 60 µg/mL of AgNPs for 24 h for five repetitions. (**c**) Average deformity rate of five repetitions after exposure to 60 µg/mL of AgNPs for 24 h. Here, Only Nano indicates the embryos were exposed to AgNPs only. Only V indicates embryos were exposed to only voltage pulses with a chamber filled with E3 media instead of nanoparticles. Nano+V indicates embryos that are exposed to both nanoparticles and voltage pulses. “*” indicates statistically significant difference (*p*-value < 0.05) when compared with control 1 (No V+ NO particle).

**Figure 8 micromachines-15-00049-f008:**
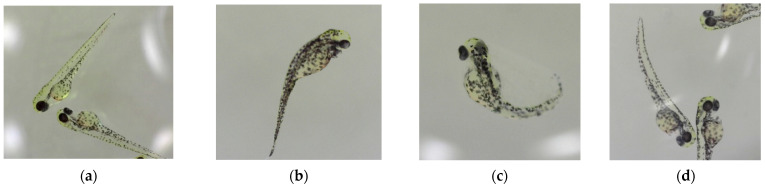
Deformities observed from exposure to 10nm AgNPs for 40 min after electroporation (**a**) Control with no deformity. (**b**) Abnormal middle body shape; (**c**) abnormal tail; (**d**) tilted tail.

**Figure 9 micromachines-15-00049-f009:**
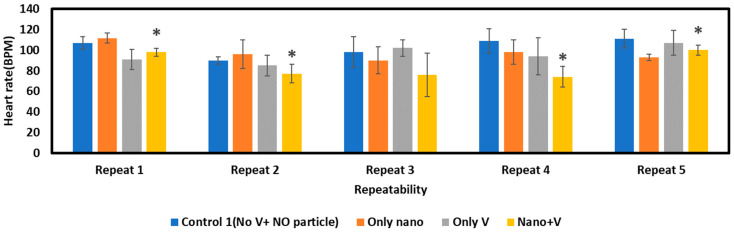
Condition of heart rate for exposure to 10nm AgNPs for 40 min. “*” indicates statistically significant difference (*p*-value < 0.05) when compared with control 1 (No V+ NO particle).

**Figure 10 micromachines-15-00049-f010:**
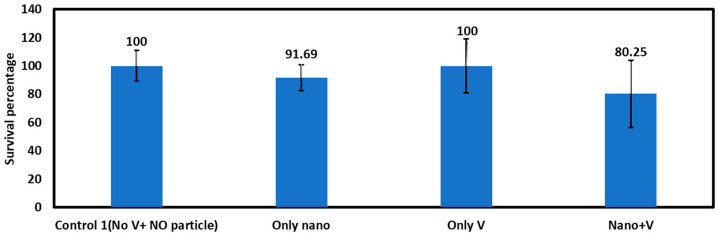
Survival rate for exposure of 10nm AgNPs for 40 min after electroporation (N = 5).

## Data Availability

The data that support the findings of this study are available within the article and the Appendix A.

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
