# Peer review of "Using Electroporation to Improve and Accelerate Zebrafish Embryo Toxicity Testing"

_micromachines, 2023, doi:10.3390/mi15010049_

Round 1
Reviewer 1 Report
Comments and Suggestions for Authors
Fish Embryo Toxicity Testing protocol based on Zebrafish embryo electroporation was developed. In general, the manuscript is well-written and might have an interest for some particular reader. I do not believe that it is the right journal because there is nothing about microdevices. Thus, 2.3 chapter does not fit the Materials and Methods. There are a lot of old and not correctly cited references (no pages and volume).
Author Response
Thank you for your comment. We apologize that the topic's relevance to the MDPI Micromachines journal was unclear. The article is submitted to the Special edition of MDPI Micromachines focusing on “Environmental Monitoring and Food Safety and Human Health In Microfluidics and Microsystem Applications”. In this contribution, we presented a microfluidic system designed for toxic screening for zebrafish embryos utilizing electroporation and included a compelling proof of concept. We believe our research aligns well with the journal’s theme and our proposed approach makes it suitable for MDPI micromachines.
We acknowledge your observation regarding Chapter 2.3. We recognize that electroporation is a well-established method and have revised section 2.3 focusing only on pertinent information essential for coherence. We refer to lines 117-124 for concise information.
I have updated the cited references and added pages and volume on pages 15 and 16.
Reviewer 2 Report
Comments and Suggestions for Authors
This paper presents the use of electroporation as a technique to improve the delivery of toxic elements inside the chorion, increasing the exposure level of the toxins at an early embryo stage (<3 hours post-fertilization). A custom-made electroporation device with the required electrical circuitry has been developed to position embryos between electrodes that provide electrical pulses to expedite the entry of molecules inside the chorion. The optimized parameters facilitate material entering into the chorion without affecting the survival rate of the embryos. The effectiveness of the electroporation system is demonstrated using Trypan blue dye and gold nanoparticles (AuNPs, 20-40 nm). Their results demonstrate the feasibility of controlling the concentration of dye and nanoparticles delivered inside the chorion by optimizing the electrical parameters including pulse width, pulse number, and amplitude. Authors also tested silver nanoparticles (AgNPs, 10 nm), a commonly used toxin which can lower mortality, affect heart rate and cause phenotypic defects. The results showed that electroporation of AgNPs reduces the exposure time required for toxicity testing from 4 days to hours. The study is interesting. In the title, what does improve and accelerate mean? Accelerate the toxicity testing sounds strange. Many figures have no scale bars.
Comments on the Quality of English LanguageSee my comments.
Author Response
Thank you for your feedback. We apologize that the title meaning was not clear. In this paper, our utilization of electroporation as a delivery method significantly enhances control over toxin delivery inside the chorion of zebrafish embryos. This method not only improves the testing process in terms of delivery methods but also accelerates the testing time from days to hours. I refer to line number 68-71 and 87-91 for clarification on the significance. The method also has the capability for implementation in large scale screening. Therefore, we believe a title emphasizing “improved and accelerate” represents the work properly. Let us know if you think further modification and explanation are required.
Query regarding the scale bar is not clear. Providing a specific number or example will help with further modification. The data presented in the figures is also provided in detail in the Supplementary document.